# Research on the Pathway of Green Financial System to Implement the Realization of China’s Carbon Neutrality Target

**DOI:** 10.3390/ijerph19042451

**Published:** 2022-02-21

**Authors:** Gaoweijia Wang, Shanshan Li, Li Yang

**Affiliations:** 1School of Economic and Management, Anhui University of Science and Technology, Huainan 232000, China; 2020201280@aust.edu.cn (G.W.); lyang@aust.edu.cn (L.Y.); 2Institute of Energy, Hefei Comprehensive National Science Center, Hefei 230031, China

**Keywords:** carbon neutral and carbon peak, green financial system, relationship between green finance and carbon neutral, prediction of CO_2_ emission, grey prediction GM (1,1) model, BP neural network model

## Abstract

To answer to global climate change, promote climate governance and map out a grand blueprint for sustainable development, carbon neutrality has become the target and vision of all countries. Green finance is a means to coordinate economic development and environmental governance. This paper mainly studies the trend of carbon emission reduction in China in the next 40 years under the influence of green finance development and how to develop and improve China’s green finance system to help China achieve the goal of “carbon neutrality by 2060”. The research process and conclusions are as follows: (1) Through correlation test and data analysis, it is concluded that the development of green finance is an important driving force to achieve carbon neutrality. (2) The grey prediction GM (1,1) model is used to forecast the data of carbon dioxide emissions, green credit balance, green bond issuance scale and green project investment in China from 2020 to 2060. The results show that they will all increase year by year in the next 40 years. (3) BP neural network model is used to further predict carbon dioxide emissions from 2020 to 2060. It is expected that China’s CO_2_ emissions will show an “inverted V” trend in the next 40 years, and China is expected to achieve a carbon peak in 2032 and be carbon neutral in 2063. Based on the results of the research above, this paper provides a supported path of implementing the realization of the carbon-neutral target of China from the perspective of developing and improving green financial system, aiming to provide references for China to realize the vision of carbon neutrality, providing policy suggestions for relevant departments, and provide ideas for other countries to accelerate the realization of carbon neutrality.

## 1. Introduction

CO_2_ emission has long been a global concern of environmental issues, in the face of increasingly severe global warming, environmental pollution and other phenomena, countries are studying and exploring how to reduce CO_2_ emission from different perspectives in order to achieve green development of economic and sustainable development of environment. The IPCC’s Special Report on Global 1.5 °C Temperature Rise (2020) stresses that to achieve the global net-zero carbon emissions—carbon neutrality goal in the middle of the 21st century, it is possible to control global warming within 1.5 °C, so as to reduce the extreme harm brought by climate change. Based on the importance of carbon dioxide emissions, as of October 2020, there have been 127 country’s governments committing to achieving carbon neutrality. Among them, Bhutan and Suriname have already achieved carbon neutrality targets, six countries including the UK, Sweden, France, Denmark, New Zealand and Hungary have written carbon neutrality targets into law, and four countries and regions including the European Union, Spain, Chile and Fiji have put forward relevant draft laws. As the largest developing country in the world, China’s rapid domestic economic development and huge energy consumption have directly led to the overall increase of CO_2_ emissions, which has brought seriously negative impacts on the environment and people’s life. Therefore, the Chinese government attaches great importance to adopting a series of policies and measures. At the Copenhagen Climate Conference at the end of 2009, the Chinese government has pledged to reduce China’s CO_2_ emissions per unit of GDP by 40–50% compared with 2005 levels by 2020. In 2016, China had signed the Paris Agreement and would work in concert with other countries to jointly answer to the global climate change issue that mankind is facing. On 22 September 2020, the Chinese government made a commitment that China would strive to achieve peak CO_2_ emissions by 2030 and achieve carbon neutrality by 2060 at the United Nations General Assembly [1]. This commitment will accelerate China’s domestic green and low-carbon transition and inspire other countries to make positive commitments and take measures of carbon reduction, which could be a strong impetus for the implementation of the Paris Agreement. Countries around the world are gradually turning carbon-neutral goals into national strategies. 

In order to achieve the goal of reducing CO_2_ emissions as soon as possible, China has adopted various policies and countermeasures in recent years. From the point of view of science and technology, they are developing clean energy, making efforts to realize the electrification of transportation, and promoting zero-carbon buildings vigorously. Around 2015, China began to focus on developing green finance from the point of view of green finance to help solve the problem of CO_2_ emissions [2]. In China’s 13th “Five-Year” plan, the concept of “green development” is introduced for the first time in China and this plan has proposed to “establish a green financial system, develop green credit, green bonds, and set up a green development foundation”. The report of the 19th National Congress of the CPC emphasizes “building a market-oriented green technology innovation system, developing green finance, and, strengthening energy conservation and environmental protection industries, clean production and clean energy industries” [3]. Green finance develops late in China, and many green financial products have even developed from scratch in recent years, which shows the efficiency of green financial development in China, the high attention the national government pays to it, and the active participation of people. Since 2015, when the CPC Central Committee and the State Council first propose the construction of the green financial system in the Integrated Reform Plan for Promoting Ecological Progress, China’s green finance has made great progress in many aspects. With the support of national policies, green loans, the issuance scale of green bonds and green investment have all increased rapidly since 2015, even showing a blowout increase from scratch. Meanwhile, the number of green enterprises listed has greatly increased and green projects have gradually emerged. Green finance has gradually developed across the country, which contributes to the realization of the carbon emission reduction target. However, the role of the development of China’s green financial system in the realization of the carbon-neutral target has not been fully reflected and China’s green financial system still has some deficiencies. Green financial development is still in the exploratory stage in China. The scale of green credit, green investment and other green products is not large enough and the kind of green financial products is not rich. Furthermore, personal and corporate carbon reduction consciousness is weak, and the way to reduce carbon emissions by using green finance is not deep, therefore green businesses and green projects fail to get strong support from investors and large loans from banks, which results in green businesses and projects struggling to grow. As enterprises are the backbone of national carbon emission reduction, green finance should play a greater role in facilitating the realization of carbon neutrality.

There is an urgent need for green finance to play an important role in the realization of the carbon-neutral target, and developing and improving the green financial system to help achieve the carbon-neutral target. 

This paper chooses green credit, green bond and green project investment as the representatives of the development of green finance. Since green finance developed in China, the government has attached great importance to promoting the development of green credit, green bonds and green project investment [4,5,6]. According to data released by the People’s Bank of China, China’s green credit balance has been only 4.85 trillion yuan as of June 2013, while it has increased to 10.6 trillion yuan as of June 2019. Before 2015, China’s green bond issuance scale has been almost zero, but it has reached 382.6 billion yuan by 2019, which is a rapid growth from scratch. Similarly, the investment in green projects has been almost zero in 2015 and before, but by 2020 it has reached 5372.9 billion yuan [7]. However, because financial transformation needs time, the development of green finance in China is also in a process of continuous exploration and rapid progress. Therefore, whether green finance has a significant effect on carbon emission reduction and how to improve green finance to promote the carbon-neutral target are urgent issues to be discussed in-depth, and they are also the issues to be studied in this paper. 

In the past, scholars have researched green finance mainly on the development prospects, theory and general research of green finance, such as the development situation and suggestions of innovation of green finance [8], green finance linked to sustainable development goals [9], limitations of green finance and new research directions [10]. At present, the research on green finance begins to combine with specific problems to emphasize its multifaceted functions, such as analysis of green technology’s upgrading strategies based on the incentive of green finance [11] and the influence of green finance on enterprise performance [12]. The scholars also study the reduction of carbon emissions from different angles. In the past, more scholars have worked on the theory of carbon neutrality, such as the methods of reducing carbon emissions [13], rules for carbon emissions trading’s impact on carbon emissions [14], the factors influencing the cost of carbon emission reduction [15], and information sharing in the use of green supply chain of carbon emissions reduction [16] and application of all walks of life in carbon emissions reduction [17,18,19]. Now, more and more scholars are combining carbon neutrality with the market to research technologies and policies of carbon reduction, such as carbon allowance trading [20], sustainable fuels and carbon capture [21], uptake or utilization technologies [22]. However, the relationship between green finance and carbon emissions has not yet been explored and no scholars have used the relevant data of green finance to predict the future trend of carbon dioxide emissions. At the same time, there is no research on how green finance can play a better role in achieving carbon neutrality. These are the contents and objectives of this paper. This paper combines green finance with the carbon neutral target and forecasts the data on carbon emissions, green credit balance, green bond issuing scale and the green project investment, and on this basis, make a study on the relationship between green finance and CO_2_ emissions. The trend of CO_2_ emission from 2020 to 2060 is predicted by using a grey GM (1,1) model and BP neural network model through MATLAB software. This paper is to study how to develop and improve the green finance of China so as to achieve the goal of carbon neutrality. It is also to provide references on formulating relevant policies for the Chinese government. Furthermore, it provides ideas for other countries to accelerate the realization of carbon neutrality and reference methods for future data prediction and research.

## 2. Literature Review

### 2.1. Research Status and Development Trend of Green Finance

Many countries currently form and develop a green financial system focused on economic growth, social security and environmental protection [23]. Appropriately strengthening green finance is an important way to effectively enhance inclusive economic growth [24,25]. In the Green Investment Report, World Economic Forum states that only investments in renewable energy, energy efficiency technologies, sustainable transport and solid waste disposal (excluding nuclear and hydropower) can be considered green investment [26]. Glomsrød and Wei (2018) point out that green finance has contributed to the growth of GDP, reduced the consumption of global coal and increased the global share of non-fossil electricity, green finance has been likely to significantly promote the low-carbon transition [27]. Green finance could help achieve sustainable development [28]. Ivanova et al. (2021) evaluate the financial market environment and listed green bonds and loans as the most promising financial instruments [23]. Green finance is a necessary condition for the optimization of enterprises’ ecological innovation structure. Green finance and environmental policies’ effective coordination can promote enterprises’ ecological innovation [29]. China’s green bond market has made great progress, rising to the top tier of global rankings. A green bond has a positive effect on enterprises’ performance, and it can promote CSR and value creation. Furthermore, a green bond can also help to attract investors to a certain extent [30]. Dikau and Volz (2021) point out that only 12% of central banks in the world have a clear mandate for sustainable development currently, and 40% of central banks are authorized to support government policies including sustainability goals [9]. However, climate risks may directly affect central banks’ traditional core responsibilities. In the future, all institutions should incorporate climate-related risks into their policy framework in order to promote financial stability and the development of green finance. Gilchrist, Yu and Zhong (2021) discuss the limitations of the emerging research on green finance, they think that the research path of green finance needs to be changed and innovated and put forward a new research path to support the goal of sustainable investment [11]. 

### 2.2. Research Status of Carbon Neutral

Globally, more than 120 countries have considered or proposed carbon neutral and there are more than 30 countries that have developed a clear plan of carbon emissions. Bhutan and Suriname have reached carbon neutral. Six countries have legislated, such as Sweden and Britain. Five countries including The European Union as a whole and Canada legislate. Fourteen countries including China and Japan have issued policies [31]. However, countries have not reached a consensus on the expression of carbon-neutral targets. Australia and New Zealand are the first countries to formulate and implement a carbon neutral system in the world, among which, Australia describes as “carbon neutral” and New Zealand describes as “zero-carbon emission system”. Subsequently, Germany states the goal as “achieving zero greenhouse gas emissions”, Norway, Denmark and other countries state it as “climate neutral”, and most countries including China state it as a “carbon neutral target” in the policies related to the carbon neutral target issued by them [32]. According to the IPCC’s Special Report on Global 1.5 °C Temperature Rise (2020), “Carbon neutrality” refers to the achievement of net-zero CO_2_ emissions when anthropogenic CO_2_ removal offsets anthropogenic CO_2_ emissions worldwide over a specified period of time [33]. 

Nowadays, research on carbon neutrality includes key problems such as carbon market design [34], carbon emissions trading system [35] and carbon leakage [36]. There are differences in countries that use nuclear power and renewables in reducing carbon emissions. Countries with large-scale nuclear attachments do not tend to lower carbon emissions, contrastingly, countries with renewables do [37]. Manta et al. (2020) start from Environmental Kuznets Curve theory to study the relationship between carbon emissions, economic growth and financial development [38]. They draw a conclusion that carbon emissions and financial development reveal a bidirectional causality, which indicates financial development may help reduce carbon emissions. The recycling technology of high energy efficiency, the technology of zero-carbon energy and the technology of negative emission make up the three key technological fields of achieving the carbon neutral target and different social subjects need different policies [1]. There are explicit ways and an implicit ways of achieving the goal of carbon neutrality. The explicit way is to conserve energy and improve efficiency, optimize energy structure and innovate technology. The implicit way is ideological innovation, for example, it is necessary to strengthen top-level design, tackle key technologies and realize the collaborative governance of reduction of carbon emission and “three wastes” [39]. Jiang et al. (2020) divide study group samples into high and low carbon in terms of the carbon emissions intensity to study the effects of green credit, green VC (Venture Capital) on carbon reduction [3]. The results show that both can significantly inhibit carbon emissions, and the “crowding-in effect” was not obvious between them. So, they insist that the commercial banks should strengthen the green credit, support the development of green VC, and enhance the enthusiasm of the whole society to reduce carbon through green finance. 

## 3. Source and Analysis of Data

In recent years, with the support of national policies, green finance has gradually developed. Since 2016, green financial events such as green credit, green project investment, green bond issuance and green company listing have increased rapidly. According to the China’s Green Financial Development Report of 2017, 2018 and 2019 released by the People’s Bank of China, as well as the CSMAR database and the CEIC database, data related to China’s green credit balance, green bond issuance and green enterprise financing since the rapid development of green finance can be obtained. From 2014 to 2019, the green credit balance of major banking institutions in China increases. Since the third quarter of 2017, the People’s Bank of China has included the performance of green credit and green bond of 24 national financial institutions into MPA assessment. In June 2018, the Central Bank decided to appropriately expand the scope of collateral for Medium Term Lending Facility (MLF), mainly including green financial bonds, corporate credit bonds which are AA+ and AA level (bonds involving small and micro enterprises and the green economy are preferred), as well as high-quality green loans. With the policy support of the People’s Bank of China in recent years, green credit has increased significantly. The loan projects are mainly energy conservation and environmental protection projects and service loans, as well as emerging sectors of strategic importance loans, as shown in Table 1. Although the green credit balance is larger overall, the growth is slow, mainly because a single green loan scale is not large enough, which limits the development of many green enterprises and projects. Commercial banks are still relatively cautious and there are more small loans. Therefore, the People’s Bank of China should give more support and encourage commercial banks to provide larger green loans so that green enterprises and projects receive timely financial support.

The green bond has become a new debt financing tool emerging in recent years, which can effectively improve the financing availability of green project financing. Green bonds were almost zero in 2015, but since seven ministries and commissions issued the Guidance on Building Green Financial System in 2016, the Shanghai stock exchange and Shenzhen stock exchange, respectively, issued the Notification of the Pilot Implementation of Green Corporate Bonds and in 2017, the China Securities Regulatory Commission issued the Guidance of Supporting the Development of Green Bonds and other relevant policies, green bonds increased as a blowout situation, hitting new highs in both number and size. At the same time, green investment began to increase in 2016 and then declined significantly in the second half of 2017. In November 2018, the Asset Management Association of China (AMAC) issued the Green Investment Guidelines (Trial), which set universally applicable normative requirements for green institutional investors requiring fund managers to conduct a self-assessment of green investment once a year and report the assessment results. Since then, green investment has grown steadily, as shown in Table 2. However, since green investment started to come into investors’ sight almost from 2016, many investors are still in a wait-and-see state for green investment, and they are worried about investing or investing with large of money. Therefore, the investment amount of green projects fluctuates, and it grows slowly. In order to develop green finance vigorously, it is necessary to support green enterprises and green projects vigorously first. Therefore, green enterprises and green projects are in urgent need of support from investors and the government. The People’s Bank of China and the government can formulate stricter standards for defining green finance, stricter disclosure mechanism and screening of high-quality green enterprises and projects so that green enterprises and projects can be green, open and transparent and investors’ investment risk concerns can be reduced.

China’s government and people have been paying close attention to CO_2_ emissions all the time. From 2000 to 2015, carbon emissions increase year by year, mainly because coal met nearly two-thirds of China’s energy needs in the past 40 years. Even so, China has been trying to transform energy structure in recent years but China’s energy structure is taken up by coal predominantly until today, which results in a lot of CO_2_ emissions undoubtedly. China’s CO_2_ emissions continue to increase after a brief drop from 2015 to 2017. According to the statistics and predictions of the CEIC database, China’s CO_2_ emissions do not show an inflection point in 2017 and may even increase until 2025. However, if we consider the carbon emission intensity which is the CO_2_ emissions per ten thousand yuan of GDP, it drops from 3.68 tons/ten thousand yuan to 2.82 tons/ten thousand yuan from 1997 to 2006, a decrease of only 0.86 tons/ten thousand yuan, while from 2006 to 2018, it drops from 2.82 tons/ten thousand yuan to 1.14 tons/ten thousand yuan which reduces by 1.68 tons/ten thousand yuan. The average decline of carbon emission intensity from 2006 to 2018 is larger than that from 1997 to 2006 significantly, which means that the efficiency of China’s carbon emission reduction is improving gradually, and the effect is significant increasingly [40]. In 2009, China pledged to reduce its carbon emission intensity by 40 to 45 percent by 2020 compared with 2005′s carbon emission intensity. In fact, at the end of 2017, China’s carbon emissions intensity already fell by 46 percent compared with 2005′s, fulfilling the target ahead of schedule. Moreover, it falls by 45.8 percent in 2018, by 48.1 percent in 2019. Chen et al. (2021) empirically tested the influence mechanism of green finance on carbon emissions by using the spatial dynamic panel model. They believe that green finance contributes to carbon emission reduction, and the development of green finance indirectly leads to the reduction of carbon emissions by reducing financing constraints and promoting green technological innovation [41]. Jiang et al. (2020) divided the research samples into high and low carbon emission groups according to the carbon emission intensity and studied the carbon reduction effects of green credit and green venture capital. The results show that both of them could significantly inhibit carbon emissions, and the development of green finance can help reduce CO_2_ emissions [3]. 

In order to verify the impact of green finance on carbon emission reduction further, this paper uses the data of green credit, green bond issuance scale, green project investment and carbon emission intensity from 2015 to 2019 to conduct a correlation test. The results of the correlation test in Table 3, Table 4 and Table 5 show that the correlation coefficients between the three green finances above and carbon emission intensity are all close to −1, and there is a strong negative correlation. At the same time, sig. of three groups of correlation test values is within the scope of the significance, indicating that green credit, green bonds issuance, green project investment have significant correlation with the change of carbon intensity and it is a strong negative correlation relationship. Therefore, since 2016, the efficiency of China’s carbon emissions reduction has been greatly improved with the help of green finance. 

On 22 September 2020, at the General Debate of the United Nations General Assembly, President Xi solemnly announced to the world that China would strive to achieve a carbon peak by 2030 and achieve carbon neutrality by 2060. This paper forecasts data on China’s CO_2_ emissions, green credit balance, green bond issuance scale and green project investment in the next 40 years and puts forward suggestions based on the forecast results. 

## 4. Analysis of Data

### 4.1. Methods of Research

#### 4.1.1. Construction of Grey Prediction Model GM (1, 1)

Grey system refers to the system which contains both known information and unknown or uncertain information. The theory of the grey system is first put forward and developed by Julong Deng (1985). The grey prediction model is an important part of the grey system. It can predict the future data in the system according to the known information. At present, it has been widely used in many sectors such as population, economy, ecology, medicine, agriculture, hydrology and disaster reduction [42]. The grey system is between white and black. Part of the information in the grey system is known and the other part is unknown, and there is an uncertain relationship among various factors in the system [43], which is very consistent with this paper. The carbon dioxide emissions and green finance-related data from 2014 to 2019 are known information. However, the predicted data from 2020 to 2060 are unknown information, and there is an uncertain relationship between carbon dioxide emissions and the development of green finance in the past few years and the future. The grey prediction GM (1,1) model does not require a large number of data samples, so it is very suitable for this paper which uses five years of green financial data for forecasting. Furthermore, this model has a good short-term forecasting effect and a simple operation process, so it is also suitable for forecasting data of 40 years in this paper [44].

Aalirezaei et al. (2021) applied the grey model GM (1,1) to predict the KPIs required by water security units in Saskatchewan, Canada, and then measured, analyzed and predicted useful indicators to motivate decisions and, and provide useful information for decision-makers to assess and predict the water security level in Saskatchewan, Canada [45]. Huang et al. (2019) believe that the ferrous metals industry is one of the largest sources of industrial energy consumption and carbon dioxide emissions in China. They used the grey prediction method to study the CO_2_ emission reduction potential of China’s ferrous metals industry from 2017 to 2030 and put forward some policy suggestions on emission reduction of China’s ferrous metals industry [46]. In this paper, the GM (1, 1) prediction model is established by using the grey forecasting method to simulate the historical data on China’s CO_2_ emissions from 2014 to 2019 and predict the CO_2_ emissions from 2020 to 2060. The forecasting principle of the GM (1, 1) model is as follows: A set of new data series with an obvious trend is generated by summing up a certain data series, a model is built to predict the growth trend of the new data series, the original data series is restored by reverse calculation with the method of reduction, and then the prediction result is obtained. The process of constructing the GM (1, 1) model for the above data is as follows:

First step: Set up a set of original data.
(1)$$x^{\left(0\right)}=(x^{\left(0\right)}(1),x^{\left(0\right)}(2),\ldots ,x^{\left(0\right)}(n))$$
*n* is the amount of data. Accumulate $x^{\left(0\right)}$ to weaken the volatility and randomness of the random sequence, and then the new data sequence can be derived: (2)$$x^{\left(1\right)}=(x^{\left(1\right)}(1),x^{\left(1\right)}(2),\ldots ,x^{\left(1\right)}(n))$$
(3)$$x^{\left(1\right)}(k)=\sum_{i=1}^{k}x^{\left(0\right)}(i),k=i,2,\ldots ,n$$

Then, the adjacent mean equal weight column is:(4)$$z^{\left(1\right)}=\left(z^{\left(1\right)}(2),z^{\left(1\right)}(3),\ldots ,z^{\left(1\right)}(k)\right),k=2,3,\ldots ,n$$
(5)$$z^{\left(1\right)}(k)=0.5\left(x^{\left(1\right)}(k-1)+x^{\left(1\right)}(k)\right),k=2,3,\ldots ,n$$

Second step: Establish first-order differential equation of one variable about t that is whitening form.
(6)$$GM(1,1):dx^{\left(1\right)}/dt+ax^{\left(1\right)}=u,$$
*a* and *u* are the coefficient to be solved, and *a* is an evolution parameter, *u* is the grey action.
(7)$$\frac{dx^{\left(1\right)}}{dt}=x^{\left(1\right)}\left(t\right)-x^{\left(1\right)}\left(t-1\right)=x^{\left(0\right)}\left(t\right)$$

Combine (6) with (7):(8)$$x^{\left(0\right)}\left(t\right)=-ax^{\left(1\right)}\left(t\right)+u$$

Third step: Solve *a* and *u*. Averaging the accumulated data to generate *B* and the constant term vector *Yn*,
(9)$$Yn=B\left[\begin{array}{c} u\\ a \end{array}\right]$$
(10)$$B=[\begin{array}{l} -z^{\left(1\right)}(2)\\ -z^{\left(1\right)}(3)\\ \ldots \\ -z^{\left(1\right)}(n) \end{array}\begin{array}{l} 1\\ 1\\ \ldots \\ 1 \end{array}],\;Yn=\left[\begin{array}{l} x^{\left(0\right)}(2)\\ x^{\left(0\right)}(3)\\ \ldots \\ x^{\left(0\right)}(n) \end{array}\right]\;$$

Use least squares to get *a*, plug *a* into the differential Equation (6), and then,
(11)$${\hat{x}}^{\left(1\right)}(t+1)=\left(x^{\left(1\right)}(1)-u/a\right)e^{-at}+u/a$$

Fourth step: Accumulate the above data inversely and recover the data, then the predicted result can be got:(12)$${\hat{x}}^{\left(0\right)}(t+1)={\hat{x}}^{\left(1\right)}(t+1)-{\hat{x}}^{\left(1\right)}(t)$$

#### 4.1.2. Perfect the Predicted Value of CO_2_ Emissions Based on BP Neural Network (Multi-Layer) Model

The BP neural network has the ability to classify arbitrary and complex patterns and the ability to map multi-function excellently. The BP artificial neural network has the thinking process of a human brain. It can get the output results through continuous learning and training, taking the relevant factors into full consideration. The BP neural network is a multi-layer feed forward network training by error’s back propagation and its algorithm is called the BP algorithm. The basic idea of the BP neural network is gradient descent, that is, using the gradient search technique to make the mean square error between the actual output and the expected output minimized. Its basic process is that the input data move forward through the input layer, hidden layer and output layer successively. On the contrary, in the back-propagation algorithm, the output error moves backwards from the output layer until it reaches the hidden layer which is at the far right of the input layer [47]. Shi (2019) selected 13 indicators and established the index system of Marine regional economic prediction by using the BP neural network. The results of comparison between the predicted value and the actual value prove that the BP neural network model has a higher fitting degree and prediction accuracy [48]. Zhang et al. (2015) used the BP neural network to predict the trend term and short-term fluctuation term respectively and take the sum of the two terms as the predicted value of the final carbon emission growth rate. They then provide theoretical guidance for the formulation of low-carbon policies [49]. A neural network with only one hidden layer can approach a nonlinear function with arbitrary precision as long as there are enough hidden layer nodes. The BP neural network used for prediction only needs three layers, and the number of output nodes can be determined in advance according to the prediction function, so the parameters affecting the prediction ability of the BP neural network are only the number of input nodes and the number of nodes in the hidden layer [50,51].

To avoid over-fitting, this paper adopts some methods to reduce the complexity of neural networks. (1) Reduce the number of neural network layers and hidden layer nodes [52]. The prediction accuracy of the BP neural network decreases with the increase of neural network layers and hidden layer nodes. Therefore, in order to limit the fitting ability of the network, the BP neural network model established has three layers and they are input layer, single hidden layer and output layer in this paper. The input layer includes three neurons (green credit, green bond issuance, green project investment), the hidden layer includes five neurons and the output layer includes one neuron (CO_2_ emissions). (2) Reduce training time. This paper adds a part that will never be zero after the loss function, then the loss function will still exist after constant optimization, which is the regularization process [53]. Then in the training process, the model needs to reduce the overall weight, which will reduce the error between the actual output value and the sample on the one hand, and the weight size will decrease on the other hand, so as to reduce the complexity of the model and fit the training data properly. So, the back propagation error function is: (13)$$E=\frac{\sum_{i}{\left(ti+oi\right)}^{2}}{2}+\frac{1}{2m}\sum_{i}w_{i}^{2}$$
(ti is the desired output, oi is calculated output, m is the number of samples, w is weight). (3) Regularization. Terminating the iteration in advance can control the size of the value parameters effectively, thus reducing the complexity of the model and limiting the capability of the network within a certain range [54]. In this paper, 40 times of training were conducted in data prediction, and the training was stopped when the error was almost 0. The times of training were few, which can avoid over-fitting caused by too much training.

### 4.2. Prediction and Analysis of Data

#### 4.2.1. Solution to Grey Prediction GM (1, 1) Model

In this paper, MATLAB software is used to establish a grey prediction model to predict, and to obtain the forecast data of China’s CO_2_ emissions from 2020 to 2060.

The comparison table of prediction accuracy is shown in Table 6:

According to the prediction accuracy level in Table 6, the accuracy of this model is tested and obtained as follows (Table 7):

Meanwhile, since −a = 0.0367 < 0.3, the GM (1, 1) model can be used for medium- and long-term prediction. 

The results of the operation and test above show that the prediction data of the GM (1, 1) model is well fitted. 

The forecast of China’s green credit balance, green bond issuance scale and green project investment from 2020 to 2060 can be predicted in the same way. The forecast results are as follows: green credit balance, green bond issuance scale and green project investment increase year by year in the next 40 years, as shown in Figure 1. However, it can also be seen from Figure 2 that in the next 5–10 years, the development rate of green credit balance, green bonds and green investment will be very slow, which is not conducive to accelerating the development process of green finance. Therefore, the development of green finance should not only be limited to developing traditional financial products, but also carrying out product innovation and developing a variety of green financial products.

Through the grey prediction model, the predicted data on CO_2_ emissions in China from 2020 to 2060 are shown in Figure 2. The blue part is the comparison between the actual value and the predicted value of CO_2_ emissions from 2014 to 2019, which almost coincide, indicating that the forecast meets relatively high requirements, and it is well fitted. However, due to the grey prediction model performing trend changes weakly and it usually only presents a single growth or decline trend, so it is necessary to improve the prediction of CO_2_ emissions further. 

#### 4.2.2. Prediction Results of BP Neural Network (Multi-Layer) Model

Using the BP neural network model to predict through MATLAB, Figure 3a is the prediction of data on CO_2_ emissions in the past. The two lines almost coincide, which indicates that the prediction results of the CO_2_ emissions by using data on green credit balance, green bond scale, the green project investment are reliable. Figure 3b shows the learning and training process of this model. It can be seen from the figure that the error of this algorithm is almost zero after 40 times of training, which indicates that the prediction data of this algorithmic is well fitted and reliable. 

Forecast data on green credit balance, green bond scale and green project investment are used as hidden neurons to predict CO_2_ emissions by the BP neural network model, that is, the green credit balance, green bond scale and green project investment are seen as influencing variables for CO_2_ emissions. Moreover, the predicted value of China’s CO_2_ emissions from 2020 to 2060 is obtained, as shown in Figure 4. 

## 5. Discussion

As prediction results of green credit balance, green bonds issuance scale and green projects investment show, they will increase continuously in the next 40 years, which can indicate that China’s green finance is going to get more attention in the future. However, the prediction results of these three parameters above do not show the volatility that is likely to occur. With the development and innovation of green finance, there will be more and more new products booming, causing the volatility or even decline of green credit balance, green bonds issuance scale and green projects investment. 

According to the final prediction result attained by BP neural network, China’s CO_2_ emissions show a trend like an “inverted V” in the future. Before 2032, CO_2_ emissions rise all the time without a downward trend, which is mainly because China has been a traditional coal country for many years, and it is difficult to completely change the way of energy consumption in a short period of time. Therefore, it does not achieve a carbon peak in 2030. After accelerating the energy transition and China’s three decades of efforts, the reduction of carbon emissions is likely to be inevitable, and the carbon emission decreases continuously from 2032. On the one hand, it reflects the possibility of realizing carbon neutrality commitment by 2030, and on the other hand, it proves the effectiveness of a series of policy measures taken by China. However, there is still a gap between the prediction result and China’s target “2030 carbon peak and 2060 carbon neutral”, which indicates it is necessary to make efforts from many aspects for China. Many factors are likely to cause this result. It may be that China’s technologies of reducing carbon emission reduction are not advanced enough, or China’s relative policies are not strong and efficient enough. Furthermore, it may also be that the incentive policy about carbon emission reduction is not significantly attractive, so it’s difficult to improve the enthusiasm and initiative of enterprises and individuals to reduce carbon emission and so on. This paper focuses on the impact of the development of green finance on the realization of carbon neutrality and puts forward some relevant policy suggestions on developing and improving the green finance system, so as to implement the realization of the carbon neutrality target of China. This paper only researches the pathway to achieve the goal of carbon neutrality from the perspective of green finance and the predicted value of the CO_2_ emissions in the next forty years are based on the relationship between green finance and carbon emissions reduction. This paper does not take other influential factors of CO_2_ emissions into account and study the pathway to achieve the goal of carbon neutrality from other perspectives. The structure of the BP neural network would change when the number of nodes in the input layer and the number of nodes in the hidden layer change [49]. Therefore, this paper selects only three input variables to predict the value of the CO_2_ emissions in the next forty years by establishing the BP neural network model, which may not achieve the optimal predictive ability of the BP neural network.

## 6. Suggestions

After analyzing the development status of green finance and combining it with China’s national conditions, this paper puts forward the following suggestions for the development of green finance based on China’s carbon-neutral target (Figure 5).

Firstly, promote the information disclosure of environmental protection enterprises and green projects. In practice, the information of environmental protection enterprises and green projects lack disclosure, and the information exchange between investors and invested parties is insufficient, leading to information asymmetry between the two sides and it is difficult to invest for investors, which directly affects the credit and financing scale of environmental protection enterprises and green projects, and even hinders the process of green carbon-reduction in China. On the one hand, the government and relevant departments should formulate unified and specific information disclosure standards for environmental protection enterprises and green projects, standardize and enrich the ways of information disclosure [55], so that investors can better understand the invested enterprises and projects and assess their risk and benefit, and then decide whether to invest or not. On the other hand, they could use financial technology to establish a shared database to realize data sharing between the two sides, so that investors can fully understand the specific situation of environmental protection enterprises and green projects and make a quick decision, while the investee can also obtain the information of investors in time, which forms an efficient communication mechanism, and improves financing efficiency. At the same time, fintech can also be applied to improve the information disclosure mechanism and create a secure and efficient database [56].

Secondly, revise the definition standard of China’s green finance to prohibit “greenwashing”. In the EU’s latest Sustainable Finance Standard, it stresses that economic activities that meet its criteria must not compromise other sustainable development goals. At present, some projects in the list of China’s green credit and green projects do not fully meet the requirements of the carbon-neutral target for net-zero carbon emissions. The green financial standards issued by China’s financial regulators and environmental protection authorities are mostly comprehensive and principled, they lack specific directories, environmental risk rating standards and identification standards. In this situation, “greenwashing” often occurs. “Greenwashing” means that enterprises whitewash their environmental protection’s image and make the public mistakenly think that they are green and environmental protection enterprises or green projects, in order to gain green credit and investment by cheating [57]. “Greenwashing” wastes a lot of social resources, increases the cost of domestic green finance, hinders the development of real environmental protection enterprises and green projects in China, and it is not conducive to China’s carbon-neutral target. Therefore, the relevant authorities need to define the standards of green finance as soon as possible to avoid “exploiting loopholes” and “greenwashing”, so that the development of a green financial system can serve the carbon-neutral goals more effectively. 

Thirdly, enhance green financial innovation and build a market mechanism dominated by high-quality carbon financial products. At present, China’s green credit, green bonds and other basic green financial products have made great progress, but the product innovation is not enough to attract more investors. First, green insurance, green credit and green investment can be combined as a special insurance-credit-investment structure. The scope of green insurance can be broadened, and the intensity of insurance can be increased to insure green projects and green financial products so that lenders and investors can more confidently support green projects [58]. Second, enrich and improve green financial instruments and develop carbon emission rights, dumping rights, energy-using rights and equity-based investment instruments. Furthermore, the government should encourage financial institutions to actively participate in the innovation of financial products with ecological compensation and emission rights as the targets by policy support and providing welfare distribution. Third, introduce private capital, social capital and international capital. Set up various private green industry guarantee funds. Encourage market participants to issue green asset securitization products and green financial securities to improve the liquidity of green financial products and spread risks [59]. Only by increasing the innovation of green financial products, improving the service system of green finance, and promoting the rapid development of green finance in China, can we help to achieve the goal of carbon neutrality. 

Fourthly, strengthen the financial incentive mechanism for green and low-carbon projects. In order to achieve the goal of carbon neutrality, it is necessary to provide more financial support for China’s green enterprises and green projects so as to make green development more extensive, more timely and deeper. Therefore, China’s government and the People’s Bank of China need to take strong measures. First, the People’s Bank of China could consider setting up a larger special lending and re-lending mechanism to support green and low-carbon projects. Second, the government can provide fiscal allocations for large-scale, high-cost and effective green projects to support enterprises to carry out green projects. For investment in green financial products, the government can provide additional benefits, and investors who invest in green financial projects can enjoy certain benefits, so as to attract investment [59]. Third, the People’s Bank of China may make a list of “green finance pass” and list the high-quality and cost-effective green enterprises and projects that comply with regulations and standards through strict scrutiny and grade them according to the scale of enterprise and project [60]. Then, both commercial banks and non-bank investors can borrow and invest directly according to the list. 

Fifth, promote the development of local pilot zones for green finance reform. At present, Guangzhou, Zhejiang and other places in China have established green finance reform pilot zones, and various provinces and cities have also introduced green finance development policies in recent years. The central and local governments should review the experience of the pilot zones established already and encourage more qualified provinces and cities in China to set up pilot zones for green finance reform, meanwhile, promote the further development of existing pilot zones. First, attach importance to the training of professionals related to green finance. At present, there is a lack of green financial talents in China, and the local areas are even less supported by high-quality talents, leading to the lack of China’s green financial innovation. Talent training can inject fresh blood, which is conducive to creating rich, novel and efficient green financial products and policies. Local governments can develop a talent incentive mechanism or set up green financial institutes and introduce welfare policies such as household registration to attract and cultivate versatile talents in finance and environmental protection [61]. Second, establish a cooperation mechanism among provinces and cities. Take the form that neighboring regions cooperate, and first-tier cities drive smaller cities to develop green finance, so as to promote green financial development process across the country. At the same time, set up green financial pilot reform zones as far as possible in the local areas and promote the joint collaboration and development between urban areas to let the green finance cover the country and to promote the realization of the goal of carbon neutrality. 

## 7. Conclusions

This paper adopts the correlation test and verifies the impact of green credit balance, green bonds issuance scale, green projects investment on CO_2_ emissions and the result proves that the development of green finance is a driving force to achieve carbon neutrality. Moreover, the BP neural network model is used to predict the past value of CO_2_ emissions and the result of a fitting test proves that it is scientific and reasonable to use the development of green finance to predict the future trend of CO_2_ emissions. This paper uses the grey prediction GM (1,1) model to forecast the future trend of green credit balance, green bonds issuance scale, green projects investment and carbon dioxide emissions from 2020 to 2060, and the results of the accuracy test demonstrate the reliability of this prediction method and prediction results. These data show an increasing trend year by year on the whole in the next 40 years. Then, the BP neural network model was used to forecast the prediction on CO_2_ emissions further, and the error was reduced to almost 0 through many times of training, which proves the rationality of this prediction method and the scientificness of prediction results. The following conclusion can be drawn: China’s CO_2_ emissions show a trend like an “inverted V” in the future. It increases year by year before 2032, and the peak in carbon comes in 2032 and from 2032 onwards, it shows a relatively steady downward trend and achieves carbon neutrality by 2063. At the same time, this paper conducts an in-depth analysis based on the prediction results and puts forward policy suggestions to implement achieve the target of the carbon neutrality of China from the perspective of developing and improving the green finance system according to the relationship between green finance and carbon neutrality.

## Figures and Tables

**Figure 1 ijerph-19-02451-f001:**
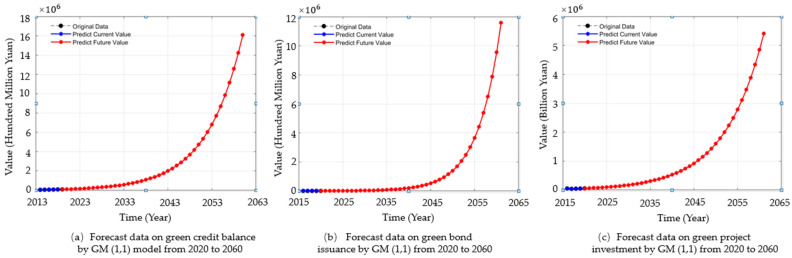
Forecast data on green credit balance, green bond issuance scale and green project investment by GM (1, 1) model from 2020 to 2060.

**Figure 2 ijerph-19-02451-f002:**
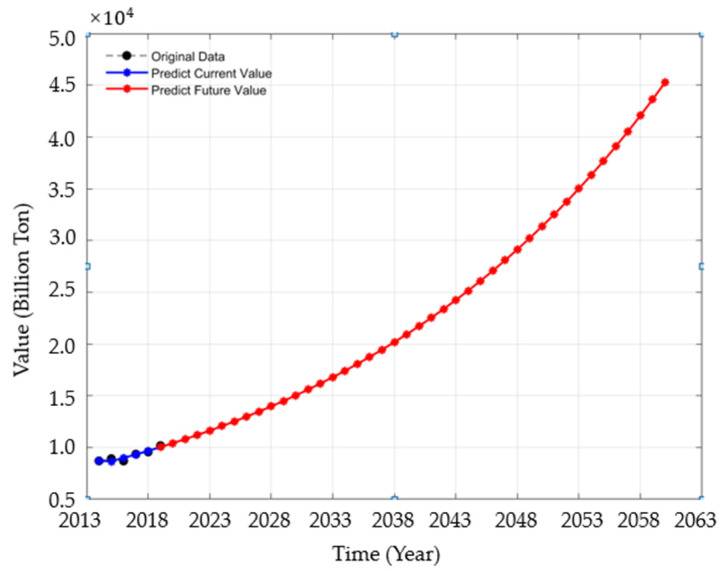
Forecast data of China’s CO_2_ emissions from 2020 to 2060 by GM (1, 1) model.

**Figure 3 ijerph-19-02451-f003:**
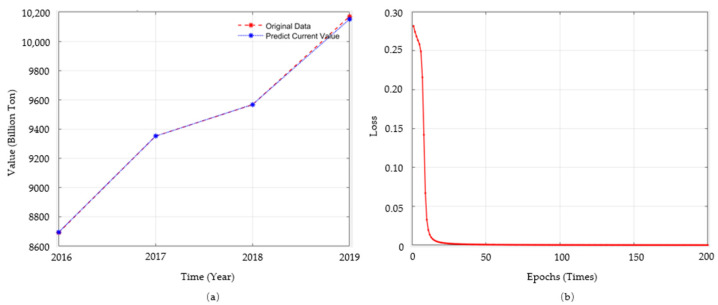
Prediction results of BP neural network (multi-layer) model. (**a**) The prediction of data on CO_2_ emissions in the past. (**b**) The learning and training process of this model.

**Figure 4 ijerph-19-02451-f004:**
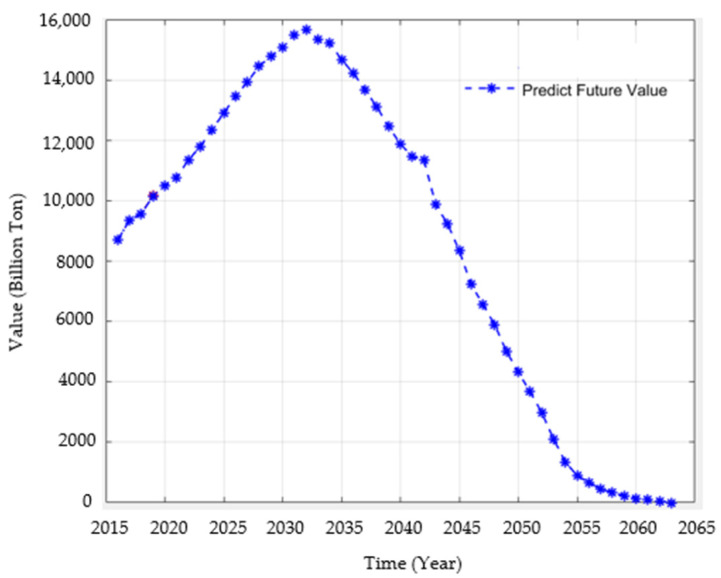
Forecast data on CO_2_ emissions from 2020 to 2060 by BP neural network model.

**Figure 5 ijerph-19-02451-f005:**
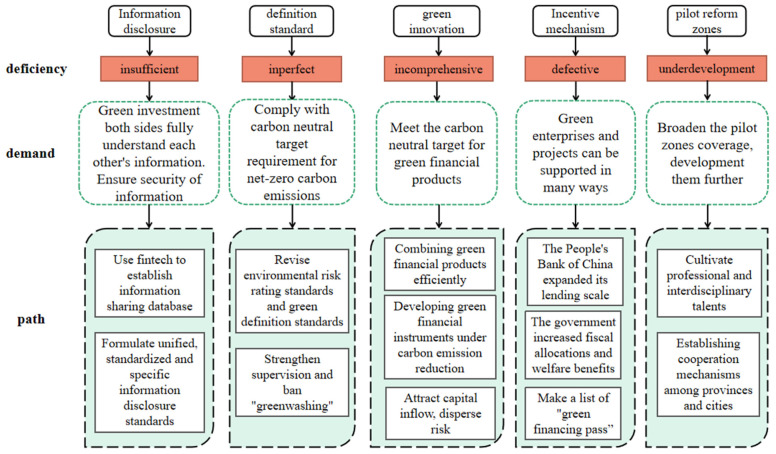
Deficiency demands and paths of green finance to help achieve carbon neutrality.

**Table 1 ijerph-19-02451-t001:** China’s green credit balance of major banking institutions from 2014 to 2019.

Deadline	Green Credit Balance (Hundred Million Yuan)	Energy Conservation and Environmental Protection Projects and Service Loans (Hundred Million Yuan)	Emerging Sectors of Strategic Importance Loans (Hundred Million Yuan)
30 June 2014	57,217.3	41,610.4	15,606.8
30 June 2015	66,361.33	49,734.66	16,626.67
30 June 2016	72,600	55,700	16,900
30 June 2017	82,956.63	65,312.63	17,644
31 December 2018	96,600		
30 June 2019	106,000		

Note: the data on energy conservation and environmental protection projects and services and strategic emerging industries loans in 2018 and 2019 are missing.

**Table 2 ijerph-19-02451-t002:** China’s green bond issuance scale and green project investment volume from 2015 to 2020.

Year	Green Bond Issuance Scale (Hundred Million)	Green Project Investment Volume (Billion)
2015	Almost zero	Almost zero
December 2016	2312	5469.9
December 2017	2497	4087.3
December 2018	2862	4708.9
December 2019	3826	5167.4
June 2020	-	5372.9

Note: Data on the size of green bond issuance in 2020 is missing.

**Table 3 ijerph-19-02451-t003:** Correlation between green credit balance and CO_2_ emission intensity.

		Green Credit Balance	CO_2_ Emission Intensity
Green credit balance	Pearson Correlation	1	−0.997
	Sig.(1-tailed)	-	0.002
	*n*	5	5
CO_2_ emission intensity	Pearson Correlation	−0.997	1
	Sig.(1-tailed)	0.002	-
	*n*	5	5

Note: Correlation is significant at the 0.01 level (1-tailed).

**Table 4 ijerph-19-02451-t004:** Correlation between green bond issuance scale and CO_2_ emission intensity.

		Green Bond Issuance Scale	CO_2_ Emission Intensity
green bond issuance scale	Pearson Correlation	1	−0.947
	Sig.(1-tailed)	-	-
	*n*	5	5
CO_2_ emission intensity	Pearson Correlation	-	1
	Sig.(1-tailed)	−0.947	-
	*n*	5	5

Note: Correlation is significant at the 0.01 level (1-tailed).

**Table 5 ijerph-19-02451-t005:** Correlation between green project investment and CO_2_ emission intensity.

		Green Project Investment	CO_2_ Emission Intensity
green project investment	Pearson Correlation	1	−0.950
	Sig.(1-tailed)	-	-
	*n*	5	5
CO_2_ emission intensity	Pearson Correlation	−0.950	1
	Sig.(1-tailed)	-	-
	*n*	5	5

Note: Correlation is significant at the 0.01 level (1-tailed).

**Table 6 ijerph-19-02451-t006:** The comparison table of prediction accuracy.

Level	Variance Ratio (C)	Small Probability Error (P)	Correlation (R)	Relative_Error_Mean	
good	C ≤ 0.35	P ≥ 0.95	R > 0.9	Accuracy level	
qualified	0.35 < C ≤ 0.5	0.95 > P ≥ 0.8	0.9 ≥ R > 0.8	excellent	<0.01
Barely qualified	0.5 < C ≤ 0.65	0.8 > P ≥ 0.7	0.8 ≥ R > 0.7	qualified	0.01–0.05
unqualified	C > 0.65	P < 0.7	0.7 ≥ R > 0.6 (satisfied)	Barely qualified	0.05–0.1

**Table 7 ijerph-19-02451-t007:** Accuracy of the model constructed.

Test Index	Test Value	Test Result
C	0.1841	Good
P	1	Good
R	0.7314	Barely qualified
Rel_Error_Mean	0.0149	Qualified

## Data Availability

Not applicable.

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
