# Peer review of "Research on the Pathway of Green Financial System to Implement the Realization of China’s Carbon Neutrality Target"

_ijerph, 2022, doi:10.3390/ijerph19042451_

Round 1
Reviewer 1 Report
The manuscript touches upon an important topic and studies it using novel machine learning models. Having said that, the analysis has several analytical drawbacks which I am not sure whether the authors could sufficiently and scientifically addressed.
- As the authors acknowledged, the production of GHG is highly complex process, which involves the atmospheric, environmental, ecological, economic, global, social, behavioral and political systems and their interactions. To what extent can we credibly model forecast it with green finance data in China alone? To what extent does green finance, a relatively new instrument, in China sufficiently influence the behaviour of the variable (I.e. China’s GHG emissions)?
- An underlying assumption of many forecasting models is that the deep parameters are temporarily stable. The authors need to justify whether this assumption can reasonably be met within the time scope of the study (i.e. the next 40 years), when we are talking about a highly dynamic economy like China.
- Trillions of yuans is a significant amount. However, how many of them turn into impactful investment, technologies and products that would contribution to the reduction of GHG?
- Regarding the model, why the Grey System was chosen? What are the analytical merits of the model? What does (1,1) refer to? If they are parameters, why were these parameters chosen?
- Relatedly, how does GM(1,1) perform relative to other simpler models? A superior performance may help to sell the models.
- If I understand correctly, given data limitation, the authors are using about 5 year’s (annual) data to forecast the next 40 years. Some people may not consider it a sound analysis.
- The inverted-U relationship is interesting. But why should we expect that? Wha are the possible reasons behind the shape of the series?
- Machine learning models like neural networks tend to overfit. This may explain the performance of the model, reported in Table 3. You may need to cross-validate the models.
- Section 5 is great. But some of the suggestions are not based on the data analysis. It would be great if the authors can bridge the gap between analysis and recommendations.
Minor issues
- Lines 64 and 68: The authors mentioned some progress and deficiencies. It would be great if the authors can elaborate them.
- Line 203: What are the 1s in B (between the closing round brackets and before the closing square bracket)
- Some materials in section 3.2 (about trends) can go to a new section right before section 3.
Reviewer 2 Report
The paper provide logic and understandable form. The cited literature is sufficient and actual. The introduction and Literature review is clear and valuable for readers. The source of the data and its commentary is sufficient. Regarding the prediction and modelling I have the comments as follows:
- The ANN model should be specified in more detail. In artificial neural networks, the complexity of the network, i.e. the number of neurons, number of layers, transfer function, etc., is very important. Especially for extrapolation this is crucial.
- There is no discussion of the reliability of the simulated data, i.e., the intercomparison of the training and test data sets. If a different network structure had been chosen, would the results have been the same? From Figure 3, it may seem that the neural network is too simple. How many layers does it have?
- The authors conclude that China will reach carbon peak in 2032, two years later than the target, and reach carbon neutrality in 2063, three years later than the target. Which input parameter to ANN carries the most weight in determining the results? Missing from the discussion here is what affects the outcome the most. Has a detailed sensitivity analysis been conducted?
The paper can be accepted, but the above comments should be addressed.
Author Response
Many thanks to the reviewer for your positive recognition and full affirmation of my manuscript. We feel much honored that the reviewer made positive comments on the significance of topic selection. The comments are valuable and very helpful for revising and improving the quality of our paper. We have studied comments carefully and have made revisions thoroughly for our manuscript in accordance with reviewers' suggestions.
Point 1: The ANN model should be specified in more detail. In artificial neural networks, the complexity of the network, i.e. the number of neurons, number of layers, transfer function, etc., is very important. Especially for extrapolation this is crucial.
Response 1:
Thanks for your valuable comments. Combining with your suggestions, we add more details about BP neural network of this paper (see lines 364-382). “The BP neural network model established in this paper has three layers and they are input layer, single hidden layer and output layer. The input layer includes three neurons (green credit, green bond issuance, green project investment), hidden layer includes five neurons and output layer includes one neuron ( emissions). Based on Sigmoid function and back propagation error function ( is the desired output, is calculated output, is the number of samples, is weight), this model constantly update and iterate the weight of each neuron and repeat training for 40 times. Finally, the error is reduced to 0.00013785.”
Point 2: There is no discussion of the reliability of the simulated data, i.e., the intercomparison of the training and test data sets. If a different network structure had been chosen, would the results have been the same? From Figure 3, it may seem that the neural network is too simple. How many layers does it have?
Response 2:
The test set data of green credit, green bond issuance scale and green project investment amount in BP neural network are predicted by grey prediction GM (1,1) model. At the same time, we use the grey prediction GM (1, 1) model to predict green credit, green bonds issuing scale and green project investment in 2015-2019 (that is the training set data of BP neural network model). The blue part of the fig. 5. is the predicted value, and red part represents the actual value. These two parts are almost overlap, indicating that fitting degree is high. In addition, this paper takes the accuracy test for the prediction model and predicted results (Table 6, Table 7.). All above show that the data of green credit, green bonds issuance scale and green projects investment predicted by the grey prediction GM (1,1) model (that is, the simulated data of the BP neural network model) are reliable.
Table 6. The comparison table of prediction accuracy.
|
level |
Variance ratio (C) |
Small probability error (P) |
Correlation (R) |
Relative _ Error _ Mean |
|
|
good |
C≤0.35 |
P≥0.95 |
R>0.9 |
Accuracy level |
|
|
qualified |
0.35<C≤0. 5 |
0.95>P≥0.8 |
0.9≥R>0.8 |
excellent |
<0.01 |
|
Barely qualified |
0.5<C≤0.65 |
0.8>P≥0.7 |
0.8≥R>0.7 |
qualified |
0.01-0.05 |
|
unqualified |
C>0.65 |
P<0.7 |
0.7≥R>0.6(satisfied) |
Barely qualified |
0.05-0.1 |
Table 7. Accuracy of the model constructed.
|
Test index |
Test value |
Test result |
|
C |
0.1841 |
Good |
|
P |
1 |
Good |
|
R |
0.7314 |
Barely qualified |
|
Rel_Error_Mean |
0.0149 |
Qualified |
The neural network model in this paper has three layers in total, among which the hidden layer is one layer and there are five neurons, which is relatively small in number, mainly in order to avoid over-fitting. BP neural network model is a complex deep learning model, but this paper uses a small data set, if the network structure is too complex, it will lead to over-fitting [50]. Therefore, reducing the number of layers and neurons can limit the fitting ability of the network.
[50] Khanlou, H. M.; Sadollah, A.; Ang, B. C.; Kim, J.H.; Talebian, S.; Ghadimi, A. Prediction and optimization of electrospinning parameters for polymethyl methacrylate nanofiber fabrication using response surface methodology and artificial neural networks. Neural Computing and Applications 2014, 25, 767-777. DOI:10.1007/s00521-014-1554-8
Point 3: The authors conclude that China will reach carbon peak in 2032, two years later than the target, and reach carbon neutrality in 2063, three years later than the target. Which input parameter to ANN carries the most weight in determining the results? Missing from the discussion here is what affects the outcome the most. Has a detailed sensitivity analysis been conducted?
Response 3:
We are so sorry that we haven’t do sensitivity analysis to test which parameter has the greatest influence on ANN. So, we add some explanation on the parameter and prediction ability of BP neural network (see lines 357-362). “A neural network with only one hidden layer can approach a nonlinear function with arbitrary precision as long as there are enough hidden layer nodes. The BP neural network used for prediction only needs three layers, and the number of output nodes can be determined in advance according to the prediction function, so the parameters affecting the prediction ability of BP neural network are only the number of input nodes and the number of nodes in the hidden layer [48].” Chen (2005) verified that the topology structure of the neural network, which is determined by the number of nodes in the input layer and the number of nodes in the hidden layer, has a great influence on its prediction ability. And the structure of BP neural network would change when the number of nodes in the input layer and the number of nodes in the hidden layer changes [49].
Both studies above show that the number of nodes in the input layer and nodes in the hidden layer have the greatest influence on ANN. This paper selects three variables: green credit, green bond issuance scale and green project investment. Therefore, there are 3 nodes in the input layer, while setting 5 nodes in hidden layer. If the selected variables increase, the nodes in the hidden layer will also increase accordingly. The prediction ability of BP neural network constructed will increase at this time [48]. In this paper, only three input variables are selected, so the optimal prediction ability of BP neural network is not achieved. We add explanation of this limitation in the conclusion and analysis section (see lines 456-460):
“The structure of BP neural network would change when the number of nodes in the input layer and the number of nodes in the hidden layer changes [49]. Therefore, this paper selects only three input variables to predict value of the CO2 emissions in the next forty years by establish BP neural network model, which may not achieve the optimal predictive ability of BP neural network.”
[48] Zhang, L. M. Artificial neural network and its application. Shanghai, China: Fudan University Press 1995, 1-92
[49] Chen, G. Analysis of influence factors for forecasting precision of artificial neural network model and its optimizing. Pattern Recognition and Artificial Intelligence 2005, 18(05), 528-534. DOI: 10.1007/BF02873109

Round 2
Reviewer 1 Report
Thanks for taking the time and efforts to improve the quality of the manuscript.
Author Response
Please see the PDF.

This manuscript is a resubmission of an earlier submission. The following is a list of the peer review reports and author responses from that submission.